# Single-Center-Single-Blinded Clinical Trial to Evaluate the Efficacy of a Nutraceutical Containing Boswellia Serrata, Bromelain, Zinc, Magnesium, Honey, Tyndallized Lactobacillus Acidophilus and Casei to Fight Upper Respiratory Tract Infection and Otitis Media

**DOI:** 10.3390/healthcare10081526

**Published:** 2022-08-13

**Authors:** Antonio Della Volpe, Pietro De Luca, Antonietta De Lucia, Francesco Martines, Piera Piroli, Luca D’Ascanio, Angelo Camaioni, Ignazio La Mantia, Arianna Di Stadio

**Affiliations:** 1Cochlear Implant and Middle Ear Unit, Santobono-Posilipon Hospital, 80129 Naples, Italy; 2Department of Medicine, Surgery and Dentistry, University of Salerno, 84084 Salerno, Italy; 3Otolaryngology Department, San Giovanni-Addolorata Hospital, 00184 Rome, Italy; 4A.O.IU.P. Paolo Giaccone–Palermo Dipartimento BiND-Sezione di Audiologia, Università degli Studi di Palermo, Via del Vespro 129, 90100 Palermo, Italy; 5Otolaryngology Department, AORMN, 61032 Fano, Italy; 6Unit of Otorhinolaryngology, Department of Medical Sciences Surgical and Advanced Technologies “GF Ingrassia”, University of Catania, 95124 Catania, Italy

**Keywords:** nutraceuticals, clinical trial, Boswellia, bromelain, upper respiratory infection, Otitis Media, auditory test, nasal spray, quality of life

## Abstract

Some nutraceuticals have been studied as supportive treatment for fighting upper respiratory tract infection and middle ear disease. Our study aims at evaluating the effect of a specific oral supplementation in the treatment of pediatric otits media. The subjects were randomly assigned by the physician (single-blinded study) to one of three groups: Control Group (CG), Treatment Group 1 (TG1), or Treatment Group 2 (TG2). Both TG were treated with Flogostop Duo (for 20 days—TG1 or 30 days—TG2) in combination with the standard treatment, while CG underwent standard treatment only. The standard treatment was nasal aerosol with Fluticasone and Mucolytic, and nasal washing with hypertonic solution. All patients were analyzed by otoscopy, impedance, fibroscopy, and pure auditory test at the baseline (T0), after 20 days (T1) and 35 days (T2). 120 children were included in the study, 40 in the CG, 40 in the TG1, and 40 in the TG2. Both TG1 and TG2 presented statistically significant differences with respect to controls in otoscopy, impedance, fibroscopy, and PTA at T2. The otoscopy improved at T2 with statistically significant value only in TG2. The impedance and fibroscopy improved at T1 both in TG1 and TG2 compared to CG. A statistically significant improvement was observed in TG2 at T2 in comparison to both CG and TG1. Statistically significant differences were observed in PTA at T2 only compared with controls. This study confirmed the efficacy of nutraceutical as supporting therapy in the upper respiratory tract infection in children. In particular, the supplement containing Boswellia serrata and Bromelain, which are molecules with strong anti-inflammatory and pain-control capacities, could add the benefit without the adverse effects which are related to NSAID use.

## 1. Introduction

Otitis Media (OM) is an inflammatory process that affects middle ear and represents the most common infection in the paediatric population. The disease, in acute form—Acute Otitis Media (OMA), makes necessary the use of antibiotics to avoid serious problems [1,2]. 15% of children between 2 and 7 years of age suffer from OM, and the prevalence of the disease decreases by age increase [1,2]. Otitis Media with effusion (OME) represents the chronic form of OM and is an important cause of hearing loss. Bad auditory functions have a negative impact on children’s speech and cognitive development [3]. There are several risk factors that predispose children to the development of OME, such as absence of breastfeeding, adenoid hypertrophy and adenoiditis, environmental smoke exposure, male gender, genetic predisposition and ethnicity, recurrent upper respiratory tract infection (URI), immunodeficiency, functional immaturity of the Eustachian tube, and gastroesophageal reflux [4].

In almost 37% of children with URI, OM can arise as complication of the infection [5]. In case of bacterial infection, OM can be persistent or relapsing despite adequate antimicrobial therapy.

Several studies have investigated the potential role of common vitamins (such as vitamin D) and oral supplements (such as echinacea, hyaluronic acid, lactobacillus, and xylitol) to reduce the number of URI and OM [6,7].

The aim of this prospective case-control study was to evaluate the efficacy of using a nutraceutical containing Boswellia serrata, bromelain, zinc, magnesium, honey, and tyndallized Lactobacillus (acidophulus and casei) to treat upper respiratory inflammation and middle-ear-related problems in the paediatric population.

The supplement Flogostop^®^ Duo contains elements that reduce local inflammation (bromelain, zinc, and magnesium), others that stimulate local and systemic immune answer (honey and tyndallized Lactobacillus), and other molecules that contrast pain, which is very common in OM (Boswellia and bromelain). Because of these characteristics, we hypothesize that this nutraceutical could be used as supporting therapy for the treatment of both URI and OM.

## 2. Materials and Methods

This study was conducted at the Department of Otolaryngology of Santobono-Pausilipon (Naples, Italy), a tertiary pediatric referral center, from January to November 2021. All procedures were approved by the local Institutional Review Board (IRB) and conducted in accordance with the ethical principles outlined in the Declaration of Helsinki. The parents of participating children signed a written, informed consent to authorize the enrollment of their child in the study.

### 2.1. General Info and Randomization

The nutraceutical we used was an oral solution (Flogostop^®^ Duo-Humana^®^ Italia S.p.A.) which, in 15 mL, contained Boswellia serrata (150 mg), bromelain (150 mg), magnesium (90 mg), zinc (5.6 mg), honey (1.7 g), tyndallized Lactobacillus acidophilus (1.5 × 10^9^ cells), and tyndallized Lactobacillus casei (1.5 × 10^9^ cells).

All patients were screened firstly by a doctor who filled in the patients’ clinical records and who did randomization using a computer; then, a different physician (without any information on the patient assigned) performed all the clinical investigations at each follow-ups. The use of two different doctors guaranteed a single-blinded analysis.

After randomization, the patients were assigned to one of three groups: Control Group (CG), Treatment Group 1 (TG1), or Treatment Group 2 (TG2). The patients in both TG were treated by Flogostop^®^ Duo in combination with the standard treatment, instead CG was only treated with standard treatment. Nasal aerosol with Fluticasone and Mucolitic, and nasal washing with hypertonic solution were the standard treatment, as defined by local guidelines [6].

All children started the treatment within two days after the first clinical consultation.

### 2.2. Inclusion and Exclusion Criteria

Inclusion criteria were age over 12 months, patients affected by OME, patients with CHL, no previous use of ventilatory tube in the last six months, general good health, and parents’ agreement to participate in the study.

We excluded children affected by acute otitis, suffering from sensorineural hearing loss (SNHL), patients with serious conditions—congenital/acquired immunologic and/or neurologic and/or metabolic diseases—and the ones whose parents did not agree with the participation in the study.

The treatment groups followed two different therapeutic schemes: TG1 received traditional treatment + 15 mL Flogostop^®^ Duo/die for 20 consecutive days, while TG2 extended the treatment for 30 consecutive days.

Patients in the CG were treated for 30 days with the standard treatment.

Three time points were identified: T0 = before treatment (baseline), T1 = 20 days after treatment with Flogostop, and T2 35 days after treatment.

### 2.3. Clinical Evaluation (Table 1)

The children were evaluated by a pediatric otolaryngologist with over 10 years’ experience. The physician evaluated the condition of patients TM using Sensera microscope (Zeiss, Oberkochen, Germany). The clinical aspect of the TM was photo-recorded and then, based on the finding, a score from 1 to 4 was assigned; 1 = TM hyperemic, 2 = TM bulging, 3 = hyperemic + bulging, and 4 = healthy TM [6]. Then, the same doctor investigated by Stortz flexible fibro-endoscopy (Stortz, Tuttilingen, Germany) and Olympus CV-170 camera (Olympus, Shinjuku, Tokio, Japan) the patients’ nose and rhynopharinx to evaluate the presence and volume of adenoid tissue and the nasal condition (i.e., hypertrophy of turbinate). The observed findings were classified using the Cassano assessment [8], with scores ranging from 1 to 4.

Powell’s health questionnaire, translated into Italian, was used to evaluate the general condition of the patients [9] at the three follow-ups (T0, T1, and T2).

### 2.4. Auditory Tests (Table 1)

Children’s hearing capacities were tested by pure tone audiometry (PTA) and MT impedance. All tests were performed by a technician with over 20 years’ experience. PTA was performed to determine the auditory threshold and the characteristics of hearing loss (i.e., SNHL or conductive hearing loss (CHL)) using a Madsen Astera (Otometrics, Taastrup, Denmark) audiometer. The impedance test was performed to evaluate the sound transmission capacity of the middle ear (tympanic membrane and ossicular bone chain) using the Clarinet Middle Ear Analyzer (Inventis, Padua, Italy) and to identify conductive hearing loss.

The children displaying a sensorineural hearing loss (SNHL) on PTA were excluded from the study. The auditory threshold of children with CHL was scored following the ASHA guidelines for hearing loss as “1” referring to mild CHL (26–40 dB), “2” indicative of slight CHL (16–25 dB), and “3” for a normal auditory threshold (−10–15 dB). Pure tone average (PTAv) was calculated, including frequencies between 500 Hz and 2000 Hz.

The findings (graphic) of impedance test were classified as: type A tympanogram, representing a normal function; type B tympanogram, indicating the presence of fluid/infection in the middle ear impairing sound transmission; and type C tympanogram, indicating negative pressure in the middle ear, as in the case of a poorly functioning Eustachian tube or obstruction of the rhinopharynx. Based on the type, a score was assigned to statistically analyze the data; 1 = type B tympanogram, 2 = type C tympanogram, and 3 = type A tympanogram.

**Table 1 healthcare-10-01526-t001:** Summary of investigations performed during the study.

**1**	*Microscopic evaluation and photo-recording of tympanic membrane* (TM), using the following score: - TM hyperemic - TM bulging - TM hyperemic + bulging - TM healthy
**2**	*Rhynopharingeal fibroendoscopy* (using the Cassano’s score to assess the adenoid tissue’s state)
**3**	*Audiological assessment* (tympanometry, pure tone audiometry,)
**4**	*Powell’ health questionnaire* (translated into Italian, to evaluate the general condition of the patients, at three follow-ups; T0, T1, and T2)

### 2.5. Statistical Analysis

One of the authors (A.D.S.) certified in biostatistics participated in the study and in the statistical design. The statistical analysis was performed by STATA^®^. One-way ANOVA was performed to analyze the score variation between the groups (TG1, TG2, and CG) at the three time points (T0, T1, and T2) for the otoscopy findings. The same test was repeated to evaluate the variance of impedance results, fibro-endoscopy, and PTA findings.

The variances in the otoscopy findings between control and treatment at T1 were measured by one-way ANOVA. The same test was done to evaluate the other three features (impedance, fibro-endoscopy, and PTA). The numeric results of health questionnaires were compared between the groups using one-way ANOVA. A Holm–Bonferroni (HB) ad hoc test was performed for each one-way ANOVA. *p* value was considered statistically significant < 0.05.

## 3. Results

A total of 120 children were included in the study: 40 were in the control group (CG) (18 female and 22 male, age average in months 52.1 ± 15.3), 40 in the treatment group 1 (TG1) (22 female and 18 male, age average in months 47.2 ± 15.7), and 40 in the treatment group 2 (TG2) (26 female and 14 male, age average in months 51.1 ± 17.5). None of the patients included in the study needed to use antibiotics during the follow-up period (Table 2).

### 3.1. Health Questionnaire

No statistically significant differences were observed at T0 between CG (14 ± 3.8), TG1 (15.2 ± 4.3), and TG2 (16.3 ± 2.8).

No statistically significant variances were observed comparing CG (11.2 ± 4), TG1 (10 ± 3.5) at T1. Instead, statistically significant differences were observed between CG and TG2 (8.3 ± 3.7) (BH: *p* < 0.05). Statistically significant variances were observed between TG1 and TG2 at T1 (BH: *p* < 0.05).

Statistically significant variances were observed at T2 (*p* < 0.0001) between CG (9.1 ± 4.6) and TG1 (6 ± 3.1) (BH: *p* < 0.01) and TG2 (BH: *p* < 0.01) (3 ± 3.5). Statistically significant variances were observed at T2 between TG1 and TG2 (BH: *p* < 0.01) (Figure 1).

### 3.2. Otoscopy

Statistically significant differences (ANOVA: *p* < 0.05) were observed at T0 between CG (2 ± 0.9) and TG1 (3 ± 1.3) and TG2 (3.9 ± 1.4).

A statistically significant difference was observed between CG (1.6 ± 0.9) and TG1 (2.1 ± 1.7) and TG2 (4.1 ± 1.4) at T1 (ANOVA: *p* < 0.01; BH: *p* = 0.003). No statistically significant differences were identified between TG1 and TG2 (BH: *p* >0.05).

At T2 statistically significant differences were observed between CG (2 ± 0.9) and TG1 (2.8 ± 1.5) and TG2 at T1 (ANOVA: *p* < 0.0001); in particular, the differences were between CG and TG2 (4.1 ± 0.3) (BH: *p* < 0.00001) and CG and TG1 (BH: *p* = 0.03) (Figure 2).

### 3.3. Impedance

No statistically significant differences were observed between CG (2 ± 0) and TG1 (1.9 ± 0.3) and TG2 (1.9 ± 0.3) at T0 (ANOVA: *p* > 0.05). A statistically significant difference was observed between CG (1.7 ± 0.5) and TG1 (1.6 ± 0.5) and TG2 (1.4 ± 0.5) at T1 (ANOVA: *p* < 0.01; BH: *p* = 0.01). No statistically significant differences were noticed between TG1 and TG2.

At T2 a statistically significant difference was observed between CG (1.5 ± 0.5) and TG1 (1.1 ± 1.1) and TG2 (1.1 ± 0.3) at T1 (ANOVA: *p* < 0.0001); in particular, the differences were between CG and TG2 (BH: *p* < 0.00001) and CG and TG1 (BH: *p* = 0.04) (Figure 3).

### 3.4. Fibroendoscopy

No statistically significant differences were observed between CG (3.1 ± 0.7) and TG1 (2.6 ± 0.6) and TG2 (2.6 ± 0.7) at T0 (ANOVA: *p* >0.05). A statistically significant difference was observed between CG (2.7 ± 0.6) and TG1 (2.3 ± 0.7) and TG2 (1.9 ± 0.8) at T1 (ANOVA: *p*< 0.0001; BH: *p* = 0.002). No statistically significant differences were observed between TG1 and TG2 (BH: *p* > 0.05).

At T2 statistically significant differences were found between CG (2.7 ± 0.5) and TG1 (1.9 ± 0.6) and TG2 (1.3 ± 0.6) at T2 (ANOVA: *p* < 0.0001), between CG and TG2 (BH: *p* < 0.00001) and CG and TG1 (BH: *p* = 0.02) (Figure 4).

### 3.5. Audiometry

No statistically significant differences were identified at T0 between CG (32.6 ± 4.2), TG1 (32.8 ± 9) and TG2 (34.3 ± 10.8) (ANOVA: *p* > 0.05).

No Statistically significant differences were observed at T1 between CG (29.1 ± 4.3) and TG1 (25.9 ± 6.1) and TG2 (23.5 ± 10.9) (BH: *p* > 0.05). Statistically significant differences were identified at T2 (*p* < 0.001) between CG (26.6 ± 5.4) and TG1 (18 ± 5.5) (BH: *p* < 0.05) and CG and TG2 (15 ± 10) (BH: *p* < 0.01) (Figure 5).

## 4. Discussion

Overall, our study showed that, compared to the traditional therapy alone, the addition to the standard therapy of a supplement with Boswellia serrata, bromelain, magnesium, zinc, honey, and the mixture of the two tyndallized Lactobacillus allowed better recovery from the upper respiratory inflammation and middle-ear-related problems, with a positive impact on patients’ wellness (Powell’s questionnaire). The supplement, despite initial efficacy after 20 days of use, reached the best of its efficacy after 35 days, as shown by the improvement of the auditory test from T0 to T2.

The use of treatment for only 20 days, although able to improve the score of fibroscopy and impedance, was not sufficient for observing clinical differences in the otoscopy findings. Extending the time of the use of the supplement allowed to ameliorate all investigated findings.

We observed the improvement of the health conditions (results of Powell‘s health questionnaire) in all children treated with the supplement. Patients improved their wellness compared to CG after 35 days (T2), independently from the extension (TG2) or not (TG1) of the use of it. This data can be interpreted as: (i) persistence of the efficacy of the supplement after withdrawal or (ii) supplement placebo effect with benefit on the patient’s wellness.

Nutraceuticals can reduce the duration of upper respiratory tract infection [10]. Della Volpe et al. evaluated the efficacy of a supplement with anti-bacterial, immunomodulating and immunostimulant molecules to ameliorate symptomatology and the clinical course of chronic Otitis Media. The authors concluded that the combination of the molecules acted at different levels, reducing the volume of adenoid tissue, improving the middle ear ventilation, and reducing the recurrence of nasal infections [6].

Although we used a different supplement, our study confirmed della Volpe et al.’s results. In fact, all patients who used the supplement presented substantial improvement of the measured outcomes (otoscopy, impedance, fibro-endoscopy, and PTA) at T2 (35 days) compared with CG. The best results were obtained after 35 days of using the nutraceutical, as shown by the improvement of the scores of each analyzed variable. Particularly, at T2, TG2 children completely resolved their OME with recovery of normal TM finding, normal tympanogram, and recovery of normal hearing thresholds with complete closure of their air-bone gap (indicative of CHL). We speculate that the improvement of the auditory outcomes was probably related to the reduction of adenoids volume as indicated by the good patency of the rhinopharynx.

However, it is important to underline that after 20 days the anti-inflammatory proprieties of the supplement allowed the improvement of fibroscopy and impedance findings. Anyway, this efficacy was not strong enough to improve the otoscopy aspect of the TM; we speculate that more time would be necessary to observe a clinical change clinically detectable by the specialist.

A study from Cardenas et al. [11] showed how the use of Lactobacillus salivarius PS7, which has a strong antagonism against otopathogens, was effective in preventing OM; in fact, the number of OM episodes during the intervention period decreased significantly (84%) when compared to the ones reported during the six-months period before the probiotic intervention. On the other hand, a double-blind, placebo-controlled trial performed by Cohen et al. [12] demonstrated that the use of probiotics or probiotics did not reduce the risk of AOM or recurrent OM, antibiotic use, or lower respiratory tract infections at 1 year.

In this current study we used Flogostop^®^ Duo which contains Boswellia serrata, bromelain, magnesium, zinc, tyndallized Lactobacillus (acidophilus and casei), and honey. Each of these elements differently acted on the inflammation which affected the upper-respiratory tract. Boswellia serrata has been used for treating chronic inflammatory diseases for centuries since it contains a series of elements (monoterpenes, diterpenes, triterpenes, tetracyclic triterpenic acids, and four major pentacyclic triterpenic acids, i.e., β-boswellic acid, acetyl-β-boswellic acid, 11-keto-β-boswellic acid, and acetyl-11-keto-β-boswellic acid) that inhibit pro-inflammatory enzymes. In particular, the acetyl-11-keto-β-boswellic acid is a powerful inhibitor of 5-lipoxygenase, an enzyme that modulates inflammation [13]. Because of these important anti-inflammatory proprieties, Boswellia serrata is commonly used for the treatment of chronic persistent inflammatory disorders [14]. As additional, Boswellia has the same anti-inflammatory and analgesic capacities of non-steroidal anti-inflammatory drug (NSAID) without side-effects on the gastrointestinal system [15]; because these characteristics the supplement with Boswellia should be perfect for treating chronic and recurrent infection/inflammation in children.

Bromelain has anti-viral and anti-inflammatory properties that support the effect of Boswellia increasing its anti-inflammatory capacities; the anti-viral effect of Bromelain ulteriorly reduces the inflammation, even stopping the viral diffusion [16].

Magnesium and zinc have both anti-inflammatory effects. Zinc has antioxidant proprieties [17] that help to reduce the concentration of reacting oxidative species (ROS); ROS contribute to the increase of inflammation and tissues damage [18] and their reduction is helpful to rapidly solve infection and inflammation. Honey [19] and tyndallized Lactobacillus [19] improve the systemic immune response acting on the gut flora and preserving the intestinal balance.

In our study, Flogostop^®^ Duo, which combines these elements, improved the local condition in the upper respiratory tract (reduction of inflammation and viral spread) and the systemic immune response (improvement of white cells action and response, reduction of ROS). Moreover, Boswellia serrata, thanks to its painkiller activity, improved the quality of life of patients without negative effects.

Limitations of the study: this study presents some limitations. As first patients in TG2 started with worsen baseline otoscopic findings than the other two groups (CG and TG1). As second, score 1 (return to normal) might be predictable in all groups. Then, we did not evaluate socio-economical differences. Finally, Powell questionnaire is a self-reported method of data collection and, although asked by a physician during the clinical consultation, it had the biases common to all these types of questionnaires.

## 5. Conclusions

This study confirmed the efficacy of nutraceutical as supporting therapy both for the treatment of upper respiratory tract infection and OME in children. In particular, the supplement (Flogostop ^®^ Duo) with Boswellia serrata and bromelain (molecules with strong anti-inflammatory and pain-control capacities) could offer the same benefit of NSAID without their adverse effects (i.e., gastric pain). We observed that the treatment had a statistically significant efficacy after 20 days of use, reaching its best efficacy after 35 days. Thanks to the lack of toxicity of the nutraceutical, it was possible to prolong the therapy. Zinc, magnesium, honey, and tyndallized Lactobacillus (acidophilus and casei) continued to improve the efficacy of Flogostop^®^ Duo by reducing the inflammation, and improving the local and systemic immune response. Based on our results, we think that the use of this supplement should be considered as adjuvant treatment for the infection of the UTI and OME in children; Flogostop^®^ Duo allows to fight the infection/inflammation and to control pain by reducing the use of NSAID.

Additional studies on a larger sample and comparing different types of nutraceuticals should be performed to confirm our results.

Clear guidelines as to the use of nutraceuticals should be defined, even considering that these elements are helpful without being harmful.

## Figures and Tables

**Figure 1 healthcare-10-01526-f001:**
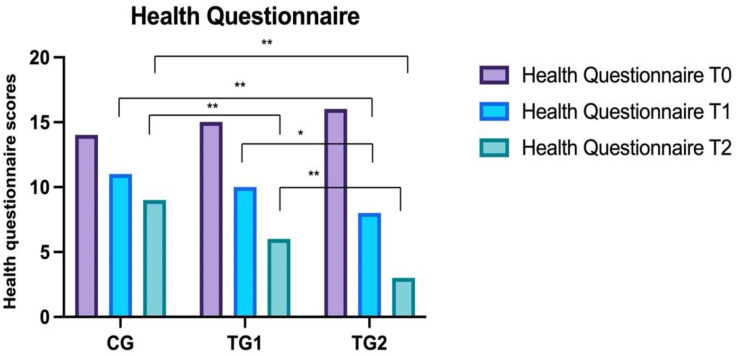
Comparison of the health questionnaire results between the three groups. Control Groups (CG), Treatment Group 1 (TG1), and Treatment Group 2 (TG2). Standard deviation “I”. The asterisk “**” indicates *p* < 0.01, “*” *p* < 0.05.

**Figure 2 healthcare-10-01526-f002:**
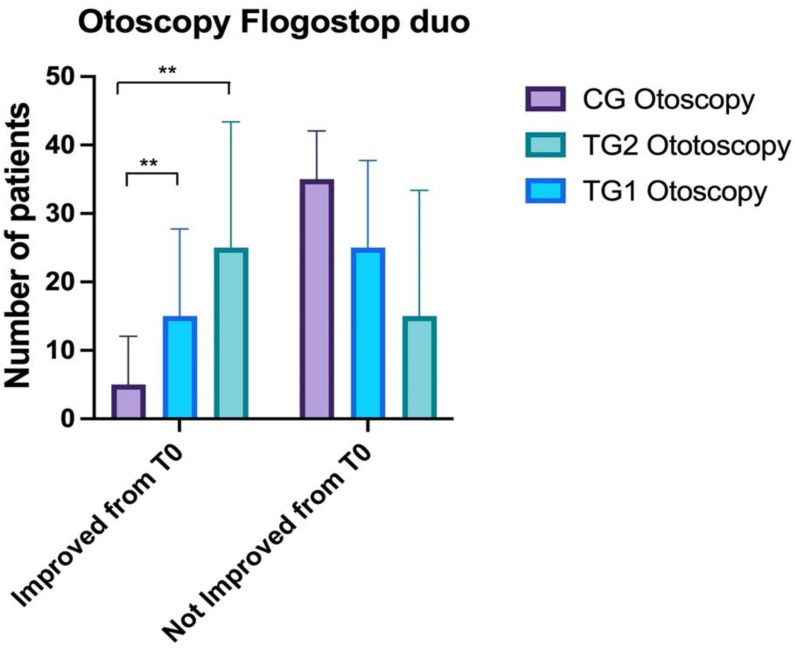
Comparison of otoscopy findings between the three groups. Control Groups (CG), Treatment Group 1 (TG1), and Treatment Group 2 (TG2). Standard deviation “I”. The asterisk “**” indicates *p* < 0.01.

**Figure 3 healthcare-10-01526-f003:**
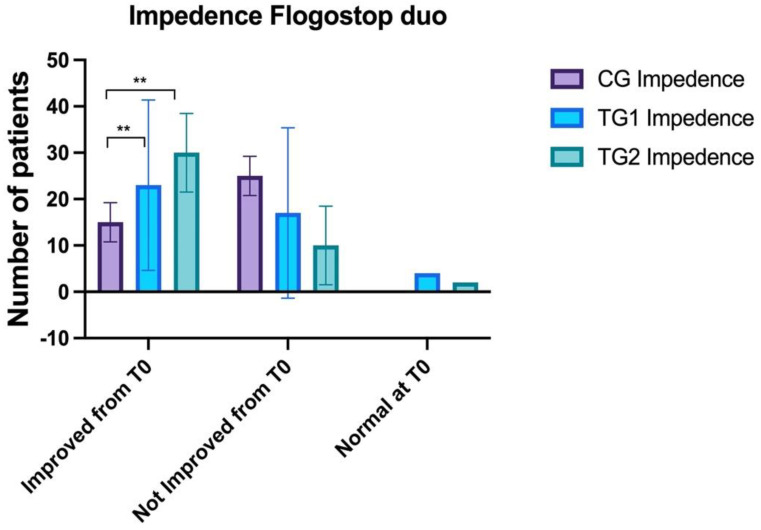
Comparison of the impedance findings between the three groups. Control Groups (CG), Treatment Group 1 (TG1), and Treatment Group 2 (TG2). Standard deviation “I”. The asterisk “**” indicates *p* < 0.01.

**Figure 4 healthcare-10-01526-f004:**
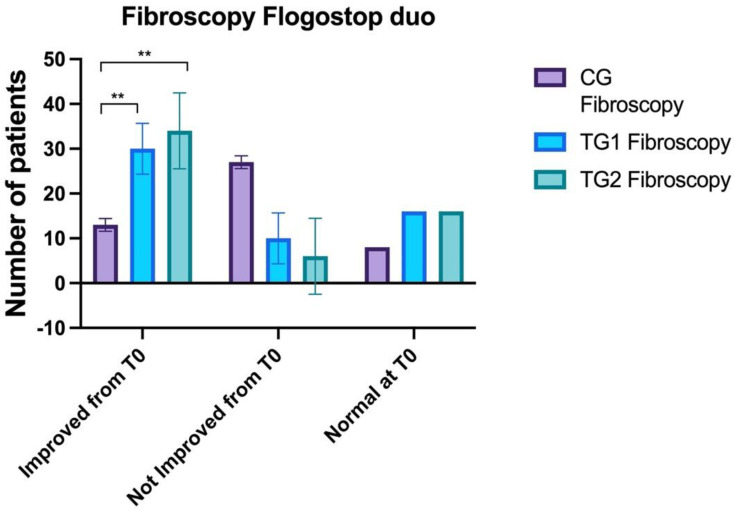
Comparison of the fibroscopy findings between the three groups. Control Groups (CG), Treatment Group 1 (TG1), and Treatment Group 2 (TG2). Standard deviation “I”. The asterisk “**” indicates *p* < 0.01.

**Figure 5 healthcare-10-01526-f005:**
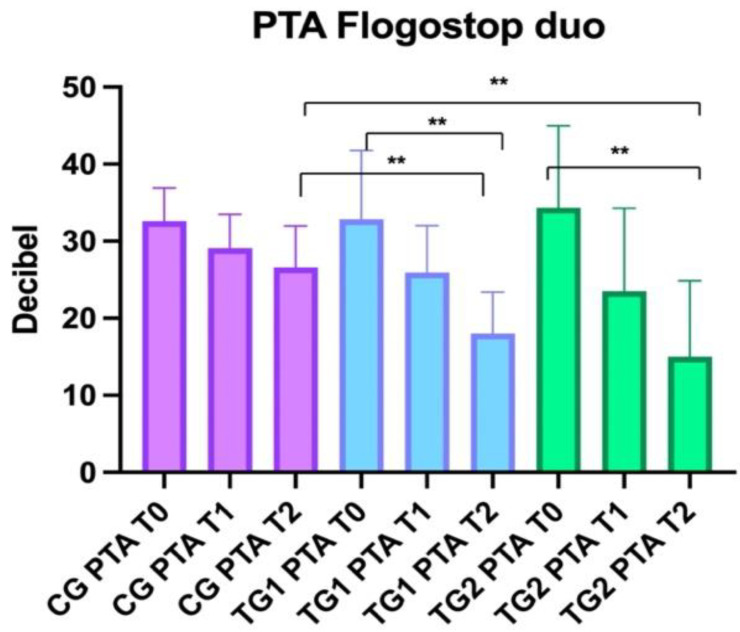
Comparison of PTA (pure tone average) between the three groups. Control Groups (CG), Treatment Group 1 (TG1), and Treatment Group 2 (TG2). Standard deviation “I”. The asterisk “**” indicates *p* < 0.01.

**Table 2 healthcare-10-01526-t002:** Demographic characteristics at the baseline.

	Number	Gender	Age (Months)	Health Questionnaire T0 *	Comorbidities
**Control Group**	40	22 male, 18 female	52.1 ± 15.3	14	none
**Treatment Group 1**	40	22 female, 18 male	47.2 ± 15.7	15.25	none
**Treatment Group 2**	40	26 female, 14 male	51.1 ± 17.5	16.3	none

*****: higher scores refer to bad health conditions.

## Data Availability

Original data are available under request to the corresponding author.

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
