# Peer review of "Single-Center-Single-Blinded Clinical Trial to Evaluate the Efficacy of a Nutraceutical Containing Boswellia Serrata, Bromelain, Zinc, Magnesium, Honey, Tyndallized Lactobacillus Acidophilus and Casei to Fight Upper Respiratory Tract Infection and Otitis Media"

_healthcare, 2022, doi:10.3390/healthcare10081526_

Round 1

Reviewer 1 Report

Thank you for the manuscript. There was some major concern.

1. It is confusing whether the author would like to investigate the product in URI or Otitis Media patients?

2. Introduction lacks the possible mechanism that FlogoStop Duo might effective.

3. Not clear on the blinding, who was blinded from the single-blind.

4. Randomization process by a physician might not be adequate to prevent bias.

5. Does the Scale for tympanic membrane evaluation validated?

6. The demographic data were not detailed to make sure that no bias was introduced.

7. The otoscopic results - the score is significant difference at baseline. The TG2 was more severe and the natural disease recovery to the score of 1 is predictable in all groups.

Author Response

Dear Editors and Reviewers,

Thank you for the thoughtful review of our manuscript and for sharing your valuable comments. We are grateful for the opportunity to submit a revised manuscript for your consideration. The critiques have allowed us to strengthen the manuscript and to address several areas requiring clarification. We have carefully read all the comments and revised the manuscript accordingly.

Below, we have responded to each comment point-by-point, presenting the reviewer critiques in regular type and our responses in bolded italics. Excerpts of revised text are also provided (indented text). In the main document, we have used track changes and red to show edits.

Thank you again for your time and insightful review of our paper.

Reviewer 1

Thank you for the manuscript. There was some major concern.

  1. It is confusing whether the author would like to investigate the product in URI or Otitis Media patients?

Thank you for this question, we investigated both. In fact, Otitis media (OM) is caused in children in particular (their tube is shorter and larger than adults) by obstruction of the Eustachian tuba due to adenoid tissue; the latter is inflamed by viral infection with consequent obstruction of the tuba and OM. However, we clarified that in the main text: “Because of these characteristics, we hypothize that it could be useful for treating both URI and OM” As additional we underlined this concept in the conclusions also “This study confirmed the efficacy of nutraceutical as supporting therapy both for the treatment of upper respiratory tract infection and OME in children”

  1. Introduction lacks the possible mechanism that FlogoStop Duo might effective.

We added a short paragraph to clarify:This supplement contains elements that reduce the local inflammation (Bromelin, Zinc, Magnesium), other that stimulate the local and systemic immune answer (Honey and tyndallized Lactobacillus) and other molecules that contrast pain, which is very common in OM (Boswellia and Bromelin).”

  1. Not clear on the blinding, who was blinded from the single-blind.

We added this relevant info. The patients were studied by two doctors; the first one, who did randomization, filled in the medical record, while a second performed the clinical investigation at each follow-up. We added a sentence to explain that. “All patients were analyzed by a first doctor who filled in clinical record and using a computer did randomization, then a second physician (without info about the assigned group) who  performed the clinical investigations at each follow-up to guarantee single-blinded analysis

  1. Randomization process by a physician might not be adequate to prevent bias.

Thanks for this note; the physician did randomization by using a computer. This missed in the previous version. We fixed the error; the sente has been corrected as “All patients were analyzed by a first doctor who filled in clinical record and using a computer performed the randomization, then anoth-er physician (without info about the assigned group) performed the clin-ical investigations at each follow-up to guarantee single-blinded analysis

  1. Does the Scale for tympanic membrane evaluation validated?

Yes, it was. We added the references of the already published work, where it was used.

  1. The demographic data were not detailed to make sure that no bias was introduced.

We added a table with demographic characteristics of the sample. (table 1)

  1. The otoscopic results - the score is significant difference at baseline. The TG2 was more severe and the natural disease recovery to the score of 1 is predictable in all groups.

We added a section with the limitations of the study that underlines these two aspects. Limitations of the study: this study presents some limitation.As first patients in TG2 started with a worsen baseline than the other two groups (CG and TG1). As second, score 1 (return to normal) might be predictable in all groups.”

Reviewer 2 Report

This is a very interesting article aiming to prove the beneficial role of the nutraceutical use (more specifically – Flogostop) in the treatment of Otitis media in children. Particularly, since the use of nutraceuticals (alone or associated with standard care) still receive a degree of reluctance or controversies, an evidence-based study to demonstrate its role is welcome.

However, some issues need to be clarified/improved.

-       how was defined the “standard treatment”? local/international guidelines? 

-       what was the place of antibiotics in the treatment of OM in this sample? None of children were in need of antibiotics treatment? were those case excluded from the sample? Do the authors maintain the same conclusions for otitis media requiring antibiotics?

-       on page 2, it is written: In almost 37% of children, OM occurs as a complication of upper respiratory tract infections (URI) and, when bacterial, can be persistent or re- lapsing despite antimicrobial therapy”. This needs a reference.  Furthermore, it may need to be checked for consistency. OM is always a complication  of an URI. Maybe the authors want to say:  37% of the URI cases may complicate with OM ?

-       on page 3 it is mentioned that “Powell’health questionnaire translated into Italian was used to evaluate the general condition of the patients”. No futher results, comments or references to this questionaire was made under Results or Discussions.

-       finally: do the authors clearly recommend that producs like FLogostop (or other nutraceuticals with similar components) to be currently used in combination. with standard treatment? is it envisaged that further guidelines would include this?

-       the authors are reminded to pay attention to the spelling and abreviations. Eg: in several instances,  the abbreviation TG 1,2. (for treatment group) is. used, in many others it is used GT 1,2. (maybe for “group treatment”?); some small spelling and English grammar errors have been found throughout/

Author Response

Dear Editors and Reviewers,

Thank you for the thoughtful review of our manuscript and for sharing your valuable comments. We are grateful for the opportunity to submit a revised manuscript for your consideration. The critiques have allowed us to strengthen the manuscript and to address several areas requiring clarification. We have carefully read all the comments and revised the manuscript accordingly.

Below, we have responded to each comment point-by-point, presenting the reviewer critiques in regular type and our responses in bolded italics. Excerpts of revised text are also provided (indented text). In the main document, we have used track changes and red to show edits.

Thank you again for your time and insightful review of our paper.

This is a very interesting article aiming to prove the beneficial role of the nutraceutical use (more specifically – Flogostop) in the treatment of Otitis media in children. Particularly, since the use of nutraceuticals (alone or associated with standard care) still receive a degree of reluctance or controversies, an evidence-based study to demonstrate its role is welcome.

However, some issues need to be clarified/improved.

  1. how was defined the “standard treatment”? local/international guidelines? 

       We used the treatment how suggested by local guidelines, the same we used in a previous published study of which we added the citation. “The standard treatment following local guidelines [5] consisted of…”.

  1. what was the place of antibiotics in the treatment of OM in this sample? None of children were in need of antibiotics treatment? were those case excluded from the sample? Do the authors maintain the same conclusions for otitis media requiring antibiotics?

Antibiotics are used only for acute otitis media in our country, these subjects were excluded from the study. We added this info in the material and method section (“Patients affected by acute otitis media were excluded from the study”). As additional, we added a sentence in the beginning of results in which we clarified that none of the patients included needed antibiotics.None of the patients included in the study needed antibiotics

  1. on page 2, it is written: „In almost 37% of children, OM occurs as a complication of upper respiratory tract infections (URI) and, when bacterial, can be persistent or re- lapsing despite antimicrobial therapy”. This needs a reference.  Furthermore, it may need to be checked for consistency. OM is always a complication  of an URI. Maybe the authors want to say:  37% of the URI cases may complicate with OM ?

We re-write the sentence. You are right it was an English form mistake. We also added the missed references n 5.: Chonmaitree T, Revai K, Grady JJ, Clos A, Patel JA, Nair S, Fan J, Henrickson KJ. Viral upper respiratory tract infection and otitis media complication in young children. Clin Infect Dis. 2008 Mar 15;46(6):815-23. doi: 10.1086/528685.

  1. on page 3 it is mentioned that “Powell’health questionnaire translated into Italian was used to evaluate the general condition of the patients”. No futher results, comments or references to this questionaire was made under Results or Discussions.

The results of health questionnaire were reported in the results section under the sub-paragraph -health questionnaire that we colored in red in this revision. As additional we also have included an additional image (now figure 1) with graph that compares the health questionnaire answer between the groups at the three follow-up.

Anyway, as you correct underlined we omitted to discuss this in the Discussion. We added a paragraph that discussed the results in this revision. „In term of improvement of health condition (results of Powell ‘health questionnaire) all children treated by the supplement improved their wellness compared to CG after 35 days (T2) and this was independent from the extension (TG2) or not (Tg1) of the use of the supplement. This data can be interpreted in two ways: i) the efficacy of the supplement persists after withdrawal or ii) the use of supplement has a placebo effect with benefit on the patient’s wellness.”

  1. finally: do the authors clearly recommend that producs like FLogostop (or other nutraceuticals with similar components) to be currently used in combination. with standard treatment? is it envisaged that further guidelines would include this.

Thank you for this remark. In the conclusions we expressed, based on the results of this study, our advice/suggestions regarding the use of nutraceuticals. We also underlined that clear guideline should be defined, even considering that these molecules are safe. 

  1. the authors are reminded to pay attention to the spelling and abreviations. Eg: in several instances,  the abbreviation TG 1,2. (for treatment group) is. used, in many others it is used GT 1,2. (maybe for “group treatment”?); some small spelling and English grammar errors have been found throughout/

Thanks, we fixed the mistakes, and we revised the text for English.